# Assessment of breast cancer surgical margins with multimodal optical microscopy: A feasibility clinical study

Mark T. Scimone[1], Savitri Krishnamurthy[2], Gopi Maguluri[1], Dorin Preda[1], Jesung Park[1], John Grimble[1], Min Song[1], Kechen Ban[1], Nicusor Iftimia[1]*

1 Physical Sciences Inc, Andover, MA, United States of America, 2 Department of Pathology, MD Anderson Cancer Center, University of Texas, Houston, TX, United States of America

* iftimia@psicorp.com

**Data Availability Statement:** All relevant data are within the manuscript.

**Funding:** Nicusor Iftimia-PI 2R44CA173998-02A1 US National Cancer Institute The funder provided

## Abstract

Providing surgical margin information during breast cancer surgery is crucial for the success of the procedure. The margin is defined as the distance from the tumor to the cut surface of the resection specimen. The consensus among surgeons and radiation oncologists is that there should be no tumor left within 1 to maximum 2 mm from the surface of the surgical specimen. If a positive margin remains, there is substantial risk for tumor recurrence, which may also result in potentially reduced cosmesis and eventual need for mastectomy. In this paper we report a novel multimodal optical imaging instrument based on combined high-resolution confocal microscopy-optical coherence tomography imaging for assessing the presence of potential positive margins on surgical specimens. Since rapid specimen analysis is critical during surgery, this instrument also includes a fluorescence imaging channel to enable rapid identification of the areas of the specimen that have potential positive margins. This is possible by specimen incubation with a cancer specific agent prior to imaging. In this study we used a quenched contrast agent, which is activated by cancer specific enzymes, such as urokinase plasminogen activators (uPA). Using this agent or a similar one, one may limit the use of high-resolution optical imaging to only fluorescence-highlighted areas for visualizing tissue morphology at the sub-cellular scale and confirming or ruling out cancer presence. Preliminary evaluation of this technology was performed on 20 surgical specimens and testing of the optical imaging findings was performed against histopathology. The combination of the three imaging modes allowed for high correlation between optical image analysis and histological ground-truth. The initial results are encouraging, showing instrument capability to assess margins on clinical specimens with a positive predictive value of 1.0 and a negative predictive value of 0.83.

## 1. Introduction

In breast cancer cases that are candidates for breast-conserving surgery, it is extremely important to achieve negative margins around the primary tumor. The margin is defined as the

support in the form of salaries for authors [MS, SK, GM, DP, JP, JG, MS, KB, NI], but did not have any additional role in the study design, data collection and analysis, decision to publish, or preparation of the manuscript. The specific roles of these authors are articulated in the 'author contributions' section.

**Competing interests:** Some of the authors [MS, GM, DP, JP, JG, MS, NI], are affiliated with Physical Sciences, Inc, which is a commercial entity interested in promoting this technology and bringing it into the clinics. This commercial affiliation does not alter our adherence to PLOS ONE policies on sharing data and materials.

distance from the tumor to the cut surface of the resection specimen. The consensus among surgeons and radiation oncologists is that there should be no tumor left within 1 to maximum 2 mm from the surface of the surgical specimen [1–3] in patients with ductal carcinoma in situ and no tumor at the inked margin in patients with invasive mammary carcinoma. If a positive margin remains, there is substantial risk for tumor recurrence [4, 5]. Published reports indicate a 20–70% rate of positive margins left after lumpectomy [6–8]. The required repeat surgery in these cases may result in potentially reduced cosmesis and eventual need for mastectomy if cancer recurrence is not rapidly detected.

Current techniques for intraoperative pathologic assessment involve touch prep and frozen section analysis [9, 10]. Touch-prep or imprint cytology allows for cytologic evaluation of the whole lumpectomy surface and has good sensitivity and specificity [11]. However, this technique is time consuming, is cumbersome, requires special expertise, and does not detect tumor foci close to the lumpectomy surface (residual cells < 2mm from the margin). Frozen section analysis is a technically challenging procedure due to the significant amount of fatty tissue found in breast specimens, which makes it difficult to freeze the tissue effectively. While the best results using this technique have been shown to reduce the rate of second operations to about 20% [12], false negatives still occur at high frequency [13]. Due to these difficulties, the vast majority of hospitals in the U.S. do not perform intra-operative margin assessment.

Clinical research approaches for intraoperative margin assessment of breast specimens include micro-CT, MRI, high frequency ultrasound, positron emission tomography (PET), and radiofrequency spectrometry. MRI and micro CT do not provide a cellular-level picture of the tissue microstructure that is necessary to assess margins accurately and therefore their sensitivity is not very high [14–16]. Ultrasound guidance for surgical margins can be useful for determining margin status in a low-cost manner, especially for women with dense breasts where other methods may not work as well. However, its reported sensitivity is rather modest (~44% in some studies) [17]. PET has shown a sensitivity of 92% to 96% and a specificity of 84% to 91% for breast margins assessment [18, 19]. However, PET is expensive, time-consuming, and invasive, which makes it not appealing for intraoperative margin assessment of breast surgical specimens. Radio-frequency spectrometry, has had limited sensitivity (60% or less) for positive margins and can only investigate relatively small surface areas (~2 mm) [20]. Limited sensitivity leads to margins left behind and thus to cancer recurrence.

Optical imaging techniques have been investigated for evaluation of physiological, chemical, and morphological changes associated with cancer, and so for facilitating margin assessment. These techniques include fluorescence imaging, Raman spectroscopy, optical coherence tomography (OCT), and photoacoustic tomography, among others [21–28]. Although most of these techniques achieved high accuracy in cancer delineation, they have not yet been translated into routine clinical practice due to some inherent pitfalls, such as reduced sensitivity, limited speed, inability to quickly cover a large tissue area, and the requirement of an expert operator [29].

However, it has been noted that some of the optical modalities provide complementary information, and thus their combined use might help to improve sensitivity, and even speed for evaluation of large surface areas of surgical resection specimens. Therefore, this paper aims to introduce a novel multimodal microscopy device that might enable more accurate detection of breast tumor margins at a reasonable speed. By combining fluorescent microscopy (FM), optical coherence tomography (OCT), and reflectance confocal microscopy (RCM), the platform provides rapid delineation and high-resolution assessment of tumor margins with minimal feedback from the operator. Briefly, using a fluorescent cancer targeting agent, FM highlights regions-of-interest (ROIs) of likely positive margins. While providing a large field of view and relatively rapid analysis of the specimens, the specificity of FM is not exceedingly high (usually prone to false positives). Therefore, combined OCT/RCM imaging is used as well

to analyze the ROI at submicron scale resolution and confirm or rule out cancer presence. RCM is used to determine if the fluorescence imaging highlighted margins are positive, while OCT is used to rapidly determine their extent in the lateral and deep aspects. In this way, the system is optimized for rapid acquisition time, keeping intraoperative utility in mind as a primary objective.

The multimodal instrument and software for image analysis developed have been evaluated on several breast tissues obtained from surgical resection specimens encountered in clinical practice. The detected margins were compared with the ground truth of histopathological examination of the imaged tissue. Twenty specimens were analyzed for this initial report. A strong correlation with histology (100%) was found in all specimens that presented positive cancer margins (8 of the total 20). The imaging of the benign specimens also correlated well with the histology. However, RCM-guided OCT was prone to false positives in a small number of specimens (two), particularly in high scattering areas of normal breast lobules.

## 2. Materials and methods

### 2.1 Fluorescence imaging agent

A contrast agent capable of highlighting areas of the specimen that have potential positive margins can be used with the proposed microscope. The main requirement is to have close to 100% sensitivity, as otherwise positive margins may be left behind. For this study, a novel contrast agent was developed by our team (see structure in Fig 1). It consists of an optically silent peptide substrate containing two (near-infrared; NIR) fluorochromes (Cy5.5), internally quenched. This agent was synthesized to track positive margins on breast lumpectomy specimens via specific urokinase plasminogen activator (uPA) cleavage.

Thus, the multifunctional peptide can be cleaved by highly expressed breast cancer enzymes, like uPA, to provide a fluorescence signal and indicate potential cancer presence at the specimen surface. Once the quenched NIR agent is cleaved, the released mono-NIR dye species become fluorescent when activated by near-infrared light. To further increase specific targeting of cancer tissue, a peptide moiety targeting Her2 or other cancer receptors may be appended, as suggested in Fig 1. A spacer (aminohexanoic acid) may be used to ensure that this peptide moiety will maintain fast uPA cleavage kinetics. However, enhancement of cancer specificity was not a major focus of this preliminary study.

The targeting agent synthesized by our team showed 90% quenching efficiency and 10X fluorescence increase after cleavage (see Fig 2). The activation time was tested as well using variable concentrations of the uPA enzyme. Based on literature data, we estimated the concentration of the enzyme that is present in malignant tissue versus benign tissue and used these numbers as a baseline for our measurements. Literature reports [30] show that the average uPA level in benign tissue is 1 ng/mg protein, while in tumor tissue is 8 ng/mg protein. Considering an average volume of tissue of 5–10 cc, 1 mm penetration depth of the agent after

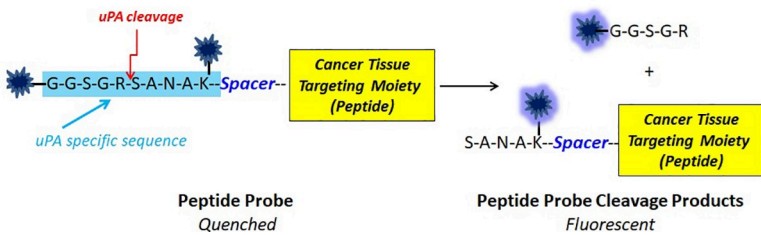

**Fig 1. Peptide cleavage and fluorescence release.**

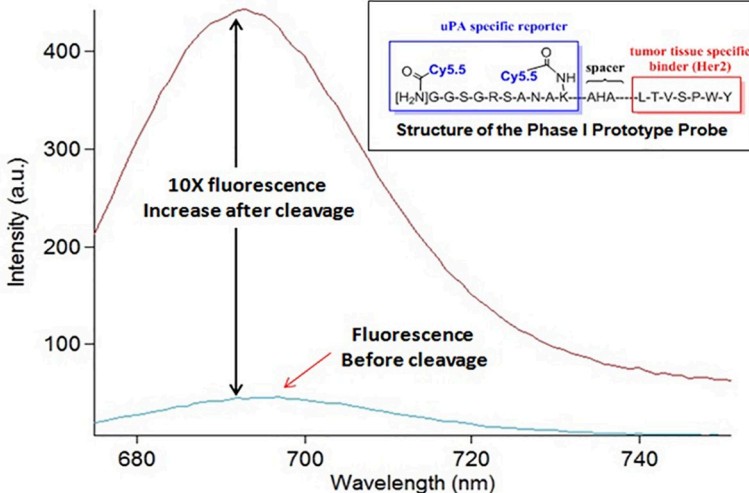

**Fig 2. Contrast agent structure and fluorescence release full protease cleavage.**

incubation, 1 g/cc density (for breast tissue), and 1uM concentration of the contrast agent in saline solution, we determined that a ratio of 1:200 between the enzyme and substrate (E:S) is representative of benign tissue, while a ratio of 1:25 is representative of tumor tissue.

However, since the tumor tissue often infiltrates benign tissue, we also considered an intermediate ratio of 1:50. Using these ratios, we performed titration kinetic activation experiments to assess the fluorescence release as a function of time and E:S ratio. The results are outlined in Fig 3. As observed, the best contrast (4.5:1 for E:S = 25 vs. E:S = 200 and 3.5: for E:S = 50 vs. E:S = 200) was obtained after only 1 minute of incubation. After that, the cleavage begins at the lower (benign) concentration of uPA, resulting in approximately half the fluorescence increase as the malignant concentration. This suggests that that the optimum incubation time will be about 1 minute. However, to enable contrast agent diffusion in the tissue, our experiments showed that a maximum of 2 minutes incubation time is a good compromise between fluorescence contrast and agent diffusivity to about 1 mm deep. For longer incubation times, the

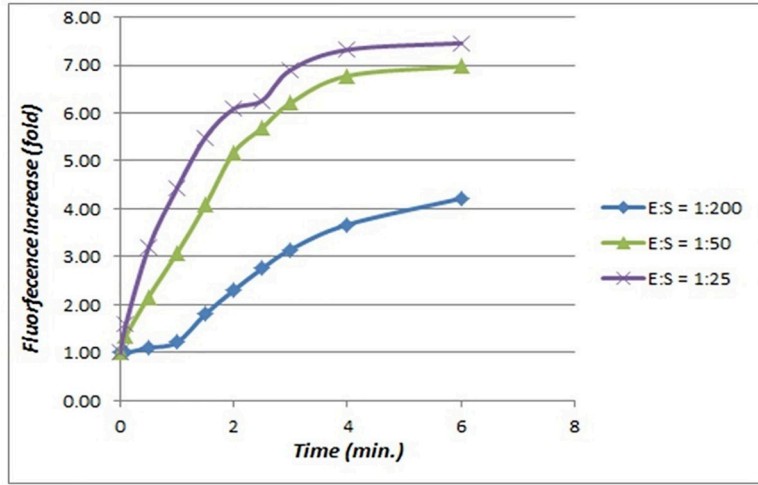

**Fig 3. Contrast agent activation kinetics.**

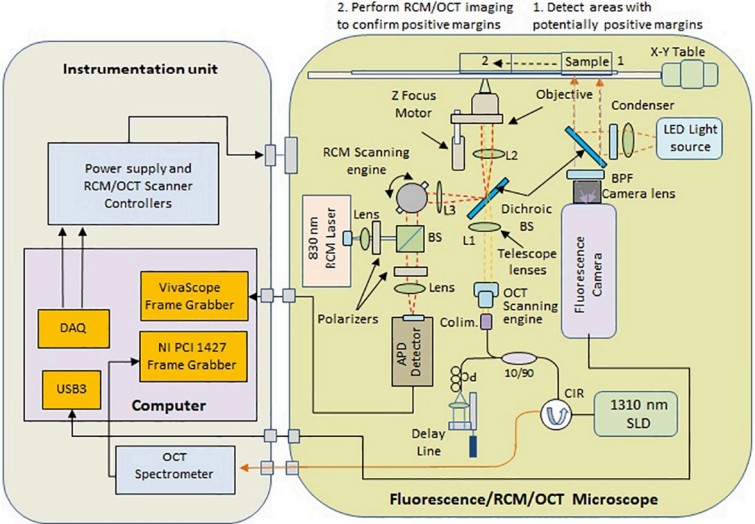

**Fig 4. Simplified schematic of the multimodal optical imaging microscope.**

smaller concentrations of the uPA enzyme present on the benign tissue start cleaving the agent, and thus degrade tumor contrast. After incubation, the unbound agent derived species are washed with saline to minimize any interference with the readout.

## 2.2 Multimodal imaging system

A simplified schematic of the optical setup is shown in Fig 4 and photographs of the instrument are shown in Fig 5.

The FM subsystem consists of a high power LED Chip light source (Model Chanzon, 380nm-840nm / 3000mA), a highly sensitive NIR camera (Model PCO Edge 4.2, PCO Tech. Inc.), dichroic mirrors and band pass filters (Chroma Technologies) to properly match the

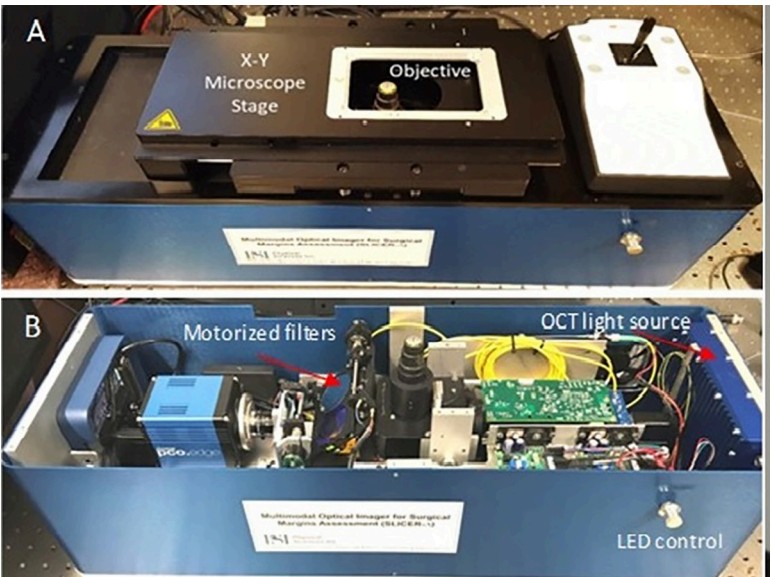

**Fig 5. Photograph of the multimodal microscope.** A. General view; B- Inside view.

excitation/emission bands of the CY 5.5 dye. The bandpass filter in front of the camera is positioned on a motorized mount, such that a brightfield image can also be taken, and overlaid by the NIR image. The field of view (FOV) of the fluorescence channel is 1.5 x 1.5 inches, which is sufficient to image many of the lumpectomy specimens. Following fluorescence ROI selection, the sample is automatically translated to be centered in the OCT/RCM objective field of view.

The RCM/OCT subsystem shares the same distal end optics (telescope lens and imaging objective), while in the upper stream the optical paths are separated by a dichroic mirror. Folding optics and telescope relay lenses are used in both the RCM and OCT optical paths such that the beam scanners can be relayed to the pupil plane of the imaging objective. The RCM subsystem is based on a typical confocal scheme. A resonant scanner (Model CR08, Cambridge Technologies, MA) and a galvanometer scanner (Model 6200H, Cambridge Technologies, MA) are used to generate a raster scan on the sample surface.

The two scanners are relayed to the imaging objective entrance pupil with two sets of relay lenses (telescope arrangement). The light from an 830 nm laser diode is directed to the scanners by a beam-splitter cube. The light returned from the sample is directed by the same beam-splitter cube to the avalanche photodiode (APD) detector. A pinhole is placed in front of the APD to reject the out-off-focus photons. Two polarizers are crossed in the emission and detecting paths to reduce the impact of sample specular reflections.

The OCT subsystem consists of a fiber optic (FO) interferometer with an optical delay line in the reference arm and a scanning engine in the sample arm. A superluminescent diode with 1310 nm central wavelength and approximately 100 nm 3dB bandwidth is used as the OCT light source. This provides a theoretical axial resolution of about 8 μm (in air), while the achieved resolution was 9 um due to imperfect balancing of the dispersion between the two arms of the interferometer. A fiber optic circulator is used to maximize light collection from the sample arm. Two galvanometer scanners (Model 6200H, Cambridge Technologies, MA), relayed to the pupil plane of the imaging objective, are used to generate a raster scan of the OCT beam on the sample surface.

Both the OCT and the RCM signals are digitized and processed by a System Control and Data Processing unit. A specially designed Vivascope frame grabber (Caliber ID, Rochester, NY) is used to digitize and process the RCM signals, enabling a display rate of 9 frames/sec. A graphical processing unit (GPU), model GTX 1060 with 1152 CUDA cores is used to expedite the OCT data processing (FFT, dispersion, and interpolation), allowing for 40 frames/sec real-time display.

## 2.3 Software for instrument control, data acquisition, and processing

The instrument control software was written in LabVIEW, employing C-based dynamic link libraries for rapid processing and easy debugging. A user-friendly graphical interface allows for instrument parameters setting, data acquisition and processing. The main steps for data collection and processing are illustrated in Fig 6.

The bright field and subsequent fluorescence images of the specimen are first collected (see Fig 6A and 6B). The user defines a region of interest (ROI) that incorporates the fluorescence image, as shown in Fig 6B, and the instrument automatically scans this area to collect spatially co-registered cross-sectional OCT images and enface RCM images (see Fig 6D and 6C–6C'). Since the size of the ROI is usually larger than the instantaneous field of view for OCT and RCM (2.2 mm for OCT and 0.75 mm for RCM), a line-strip imaging approach is used. The number of strips and their length is automatically calculated by the software as a function of the ROI size. An overlap between adjacent strips is user-defined to enable the cross-correlation of the adjacent strips for full area image stitching.

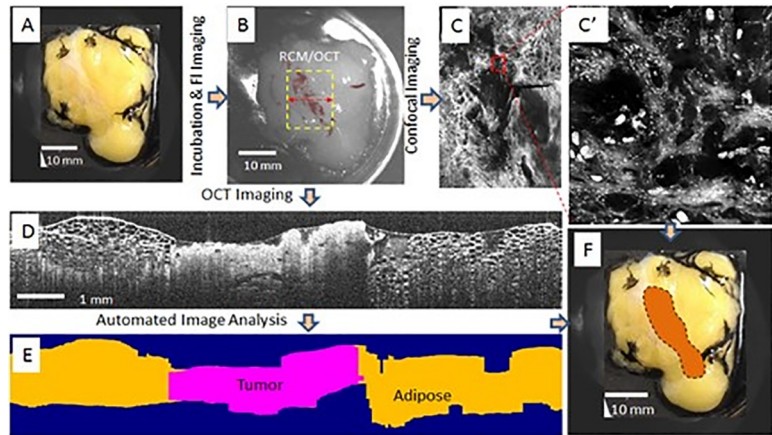

**Fig 6. Main steps for data acquisition and processing.** A: Bright field image of the specimen; B: False-color fluorescence imaging taken after specimen incubation with the contrast agent; C and C': RCM images and zoomed in region shown in red box, respectively., D: OCT image of one B-scan; E automated segmentation of the OCT data through full volumetric image (one cross section shown here); F- Overlay of the positive margins on the specimen picture.

Note that the RCM imaging over the entire area highlighted by FL imaging is not performed, as it would take too long on large areas. Only small regions of interest (ROIs) are used to determine if areas indicated by fluorescence imaging are cancer positive or not. Once these areas are confirmed, OCT imaging and segmentation is performed to determine the extent of the positive margins. Tissue-type classification is based on the analysis of the RCM image. With minimal user feedback (assignment of distinctive tissue-types on 2–3 locations of the RCM image), an automated algorithm (see Section 2.2) analyzes the collected images, determines tumor lateral and depth spreading on the OCT image (see cross-sectional tissue-type assignment on Fig 6E), and overlays the tumor margin on the specimen surface image (see Fig 6F).

## 2.4 Semi-automated software algorithm for tissue-type assignment

Following the collection and analysis of 3–5 RCM ROIs for each tissue type in different locations of the fluorescence highlighted area, such that both cancer and benign tissues are represented as determined by the pathologist, the overlapping enface OCT images are automatically assigned to the cancer and benign tissue types and used as training sets for the OCT image analysis algorithm. Representative training images are shown in Fig 7A.

Initial singular value decomposition and principal component analysis indicated that tissue slope and standard deviation of roughness are the most discriminating features.

The standard deviation of roughness and tissue slope were the most discriminating features, and their calculation is detailed further. Remarkably high frequency information was smoothed using a Gaussian kernel. Then, to calculate the roughness, the waviness profile was first determined and subtracted from the filtered image. Finally, a standard deviation filter was convolved with the roughness image. Tissue type assignment based off standard deviation of roughness was determined using values from the training image sets. The tissue slope was computed through the Prewitt image gradient method. Again, tissue-type assignment based on tissue slope was determined using values from the training image sets. All analysis was done in MATLAB, where the analysis and plotting take approximately 1 to 2 minutes, depending on the image size. The processing duration can be further shortened by implementing this code in C++.

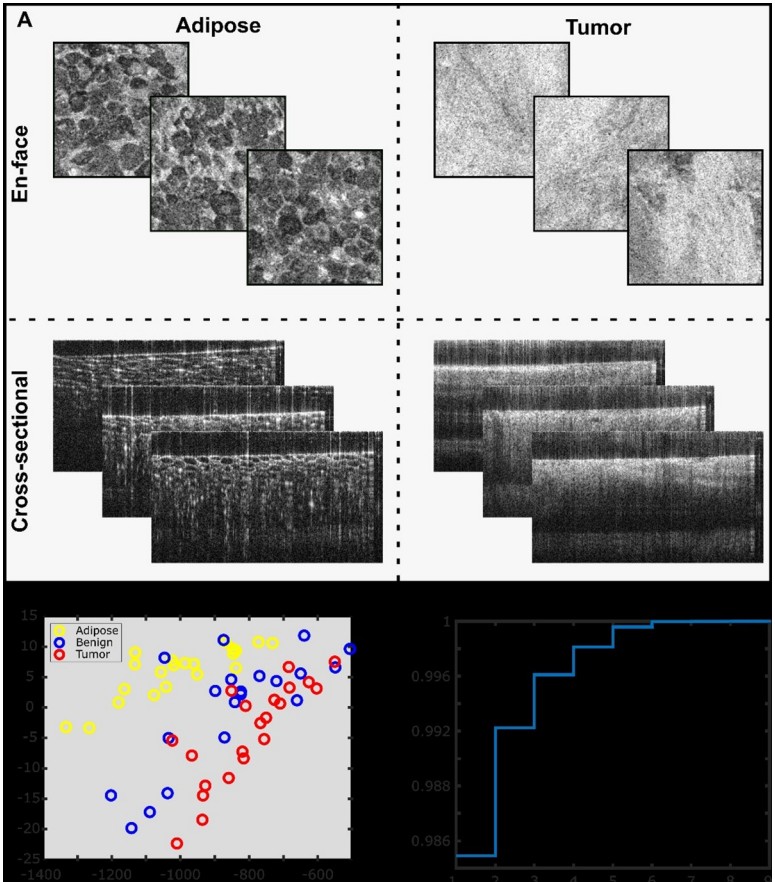

**Fig 7. Simplified representation of the automated algorithm for tissue differentiation. A.** Representative en-face and cross-sectional (b-scan) training images for adipose tissue and tumor tissue (benign representative images not shown for brevity). **B.** Principal component analysis of 9 spatial texture features shows some clustering in the 2 strongest components, but also considerable overlap. **C.** Singular value decomposition cumulative stair function showing that 5 or fewer parameters should enable strong classification capability.

Three-dimensional tissue assignment (in cross-sectional images, and in different z-planes of en-face), required depth adjusted image processing. That is, signal drop in depth was calculated for each image so that tissue-type assignment thresholds could account for it. Additionally, in cross-sectional images, the top and bottom surfaces needed to be segmented. The top surface is highly reflective and thus can be segmented by a peak-finding algorithm. The bottom (or maximum signal depth) was segmented by a histogram-based region-property analysis.

## 2.5 Tissue specimen collection and preparation

Breast surgical specimens were provided by MD Anderson Cancer Center. Twenty de-identified specimens were acquired from breast cancer patients undergoing surgical procedures (both lumpectomy and mastectomy) under the IRB approved protocol PA14-1036. The de-identified specimens utilized in the study were collected from breast surgical resections between 7/11/2019 and 11/13/2019 with waiver of informed consent. The MD Anderson Cancer Center Institutional Review Board approved the procurement of tissue for the study. Freshly collected specimens were incubated with the contrast agent for a maximum of 2 minutes, washed with saline twice, and placed on the microscope tissue holder. The specimens were slightly pressed against the microscope window facing the imaging objective, to

## 3. Results

The summary of the imaging and histopathology findings is presented in Table 1. Of the 20 specimens analyzed, histology showed positive cancer margins on 8 specimens and negative margins in the remaining 12. Fluorescence imaging showed positive margins in 10 of the 20 specimens, so all specimens with positive margins were identified. This is particularly important as no positive margins were missed. Fluorescence imaging was used to further guide RCM-OCT imaging. RCM imaging was used as input to the OCT segmentation and has correctly identified most of the tissue types, as it provided the resolution close to histopathology. Some errors were encountered in areas of normal lobules, which showed similar appearance to the cancer areas. OCT imaging correctly identified all 8 specimens with positive margins and showed false positive margins on other two specimens (mainly on areas of normal lobule locations, which showed similar scattering and image patterns like the cancer areas).

Five representative examples of margin assessment on breast surgical specimens are shown in Figs 8–11. The five cases were chosen to show distinct types of specimens that might be found during surgery: 1) specimens with positive margins all around, 2) heterogeneous specimens with infiltrating tumor margins, 3) negative margin specimens that visually contain only adipose tissue at its surface, 4) heterogeneous benign specimens that after visual and tactile examination show areas similar to cancer, and 5) specimens with small deeper cancer foci, which may been missed by histopathology if the cuts are very close to specimen surface.

The first case (see Fig 8) is that of a fragment of a mastectomy specimen, which presented nearly complete positive margins, as it was not collected with the purpose of preserving positive margins. Imaging was performed on a single surface, since visually the specimen showed similar tissue nature all around. The goal of imaging this specimen was to determine the optical imaging/algorithm capability to correctly estimate cancer presence on the entire specimen surface, as indicated by histology. Indeed, as observed in the fluorescence image (Fig 8B), nearly the entire specimen surface showed enhanced fluorescent signal relative to the background, suggesting uniform cancer presence at the specimen's surface. Cancer presence on over 80% of the specimen surface was confirmed by histology (see the red area annotated by the breast pathologist in Fig 8C). The enface segmented OCT image taken near specimen surface (see Fig 8D), as well the subsurface cancer presence, over a depth of approximately 1 mm, as shown projected on Fig 8G, confirm a similar cancer spreading area.

The second case, shown in Fig 9, is that of a small surgical specimen with positive margins (cancer infiltrating at the specimen's surface). A fluorescence signal was noticed on one of the faces of the tissue specimen, as shown in Fig 9A. However, as confirmed by histology, the fluorescence area overestimated cancer presence, as the signal appears to be heightened in areas of adipose as indicated by the histopathology's-annotated areas in the histology slide (see Fig 9C). The segmented enface OCT image taken near specimen surface (see Fig 9D), shows reasonable

**Table 1. Summary of imaging and histopathology findings.**

| 20 Specimen study | Fluorescence imaging | RCM guided OCT segmentation | Histopathology |
|---|---|---|---|
| **Positive margins specimens** | 12 | 10 (2FP) | 8 TP |
| **Negative margins specimens** | 8 | 10 (0FN) | 12 TN |
| **RCM-guided OCT predictive values: PPV = 1.0; NPV = 0.83** | | | |

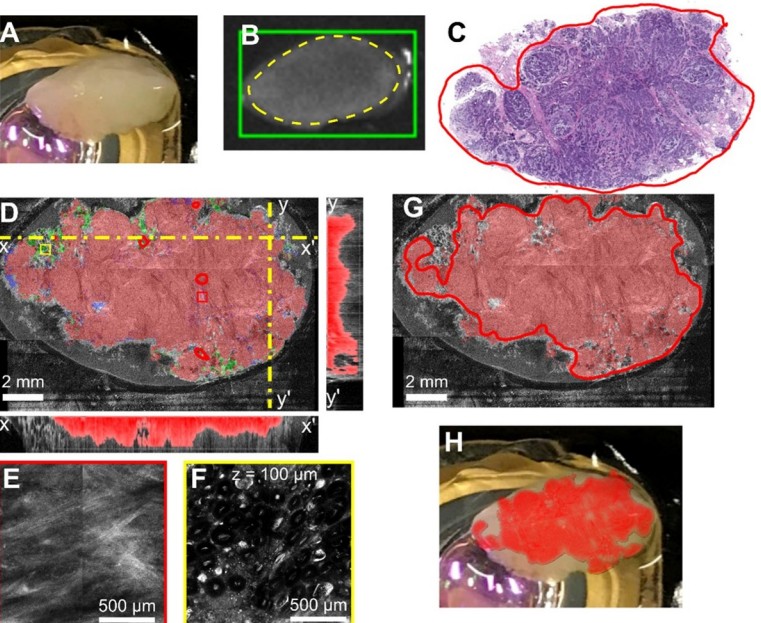

**Fig 8. Multimodal optical imaging findings on a small surgical resection specimen (~14 mm x 8mm). A**. Camera image of the specimen on the microscope stage. **B**. Fluorescence imaging, where the green box shows the selected imaging ROI in the software, and the yellow-dotted line shows the area where there is higher fluorescence from the contrast agent. **C**. The histology slide, where red indicates invasive ductal carcinoma. **D**. Segmented enface OCT image showing cancer margins with corresponding X and Y cross-sectional images. Small red and yellow boxes show region of RCM images in panels E&F. Colored overlays shows algorithmic segmentation of tissues: tumor (red), adipose (green), and benign (blue). The cross-sectional images have segmented tumor overlaid in red. **G**. Projection of the cancer areas from the multiple OCT slices over a depth of approximately 1 mm. **H.** Overlay of the positive margins on the surgical specimen.

overlap with histology relative to cancer presence. Hhistology preparation led to loss of fatty areas towards the top of the specimen as the processing of the fatty tissue for histopathological sectioning is usually difficult. This is clear when comparing the histology to the OCT (Fig 9D and 9G) and camera image (Fig 9A). The cross-sectional segmentation and Fig 9G show that the area of cancer spreads as depth increases. This highlights the need for subsurface OCT imaging and volumetric segmentation. The RCM zoomed-in panels (Fig 9E and 9F) show the structure in a tumor and fat area, respectively.

The third example is that of a surgical specimen free of positive margins (see Fig 10), as indicated by the histology image (Fig 10C), which is shown to have only a benign breast lobule (yellow) and normal duct (green). The fluorescence imaging, however, indicated a small area of potentially positive margins (see Fig 10B, red asterisk). Confocal imaging of the fluorescence-highlighted area has indicated the presence of a benign lobule. Although in real clinical practice additional OCT imaging would be unnecessary when RCM indicates the absence of the positive margin, in this case we performed OCT imaging to confirm that the segmentation algorithm assigns this area to the correct tissue class. The enface OCT image (Fig 10D) of the specimen surface clearly shows the presence of fat cells on over 90% of the specimen surface, as well as of a small area of a benign breast lobule (see blue area on the right upper area of Fig 10D), as confirmed by RCM imaging (Fig 10E). Fig 10F confirms the presence of the fatty tissue on most of the specimen's surface, as indicated by the segmented OCT image (Fig 10D).

The fourth case is that of a heterogeneous specimen, where multiple tissue types are apparent by eye (Fig 11A). After incubation with the fluorescent contrast agent, a small region in the

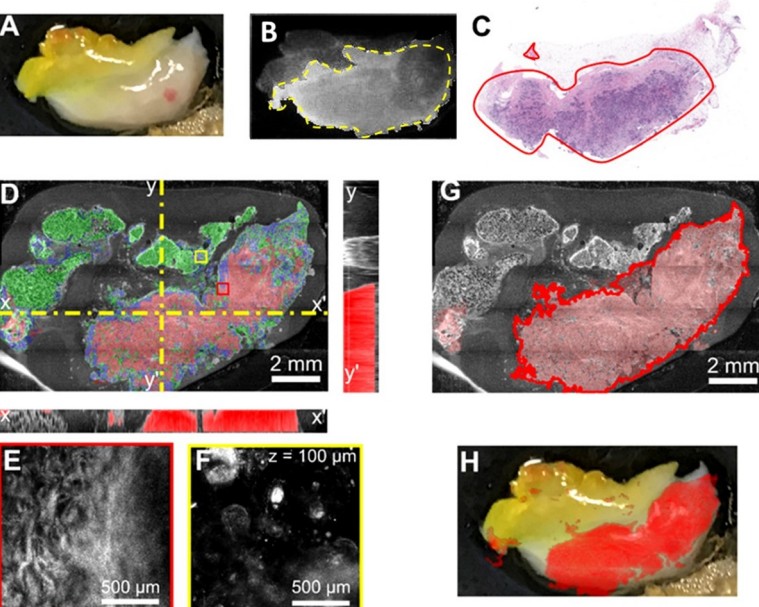

**Fig 9. Example of positive margins identification on a small lumpectomy specimen. A.** Camera image of the specimen on the microscope stage. **B.** Fluorescence imaging, where the red box shows the selected imaging ROI in the software, and the yellow-dotted line shows the area where there is higher fluorescence from the contrast agent. **C.** The histology slide, where red indicates invasive ductal carcinoma. **D.** Segmented enface OCT image with corresponding X and Y cross-sectional images. Dotted-yellow line shows the plane of the cross-sectional images. Colored overlays show algorithmic segmentation of tissues: tumor (red), adipose (green), and benign (blue), Colored boxes show region of RCM images in panels **E-F**. The z-depth shown in **F** corresponds to the imaging depth for all RCM panels **G)** Projection of the cancer areas from the multiple OCT slices over a depth of approximately 1 mm shows larger predicted tumor areas at depth than in panel D. **H.** Overlay of the positive margins on the surgical specimen.

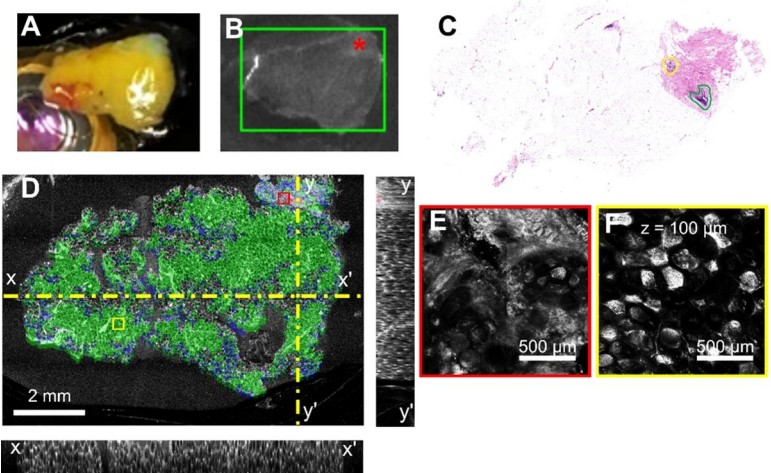

**Fig 10. Example of negative margin identification on a small lumpectomy specimen. A)** Camera photo of the specimen on the microscope stage. **B)** Fluorescence imaging seems to indicate positive margin presence (red asterisk). Green square indicates the imaging ROI selected in the software. **C)** Annotated histology indicating a benign breast lobule (yellow) and normal duct (green). **D)** Enface OCT image with corresponding X and Y cross-sectional images. Dotted-yellow line shows the plane of the cross-sectional images. Red and Yellow boxes show region of RCM images in E and F panels. Colored overlays show algorithmic segmentation of tissues: tumor (red), adipose (green), and benign (blue). **E)** RCM image of lower part of the normal duct area. **F)** RCM image of adipose tissue. z-depth of RCM image is consistent at 100 μm.

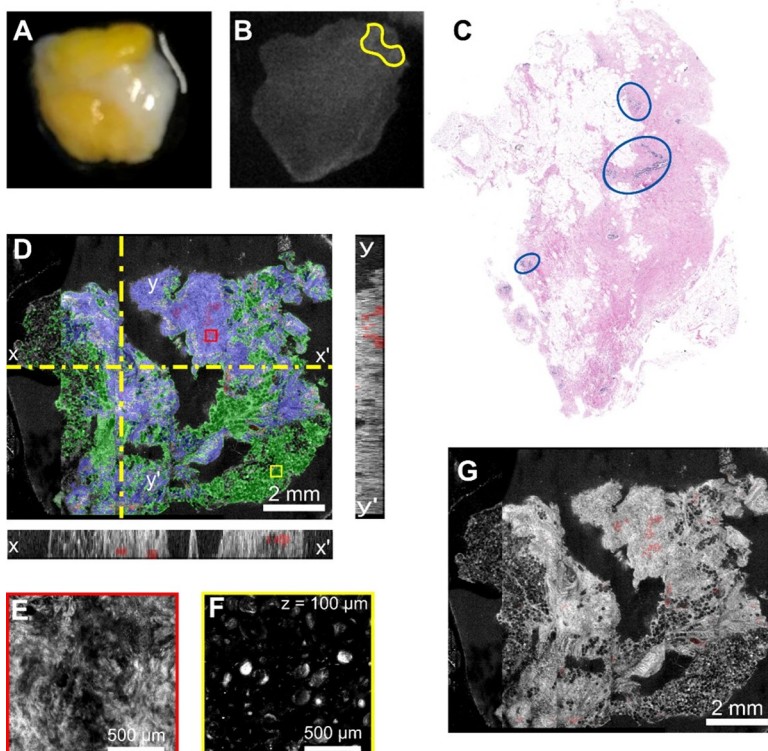

**Fig 11. Example of negative margin identification and false positives on a heterogeneous benign specimen. A)** Camera photo of the specimen on the microscope stage. **B)** Fluorescence imaging seems to indicate an area of tumor in the top-right corner (yellow). **C)** Annotated histology indicating benign breast lobules (blue). **D)** Enface OCT image with corresponding X and Y cross-sectional images. Dotted-yellow line shows the plane of the cross-sectional images. Red and Yellow arrowheads show the region of RCM images in E and F panels. Colored overlays show algorithmic segmentation of tissues: tumor (red), adipose (green), and benign (blue). Colored boxes show region of RCM images in panels E & F. **E)** RCM image of lower part of the normal duct area. **F)** RCM image of adipose tissue. z-depth of RCM image is consistent at 100 μm. **G)** Enface OCT image with only the false positive algorithmic tumor predictions shown.

corner showed increased signal (Fig 11B), and thus warranted in depth analysis. The histological annotation shows the presence of multiple benign breast lobules (Fig 11C, blue circles). Again, these structural features contribute to small false positive tumor identification in the algorithmic segmentation. This is shown in Fig 11D and 11G. However, the RCM panel in Fig 11E was interpreted as benign by the histologist, showing how multiple imaging modes can inform a more accurate classification of the tissue.

The fifth case is that of a heterogeneous specimen, where multiple tissue types are apparent by eye (Fig 12A). After incubation with the fluorescent contrast agent, several regions showed increased signal (Fig 12B), and thus warranted in depth analysis. The histological annotation shows the presence cancer areas on the left and bottom areas of the specimen (red-annotated areas) (Fig 12C). However, this histology slide seems to be been taken close to the specimen surface, as cancer foci in other areas of the specimen were identified by optical imaging. The RCM panels in Fig 12F and 12G, indicate areas of admixed fibrous-tumor tissue (F) and fibrotic tissue (G). OCT image segmentation over the 1 mm depth has correctly indicated positive margins in areas indicated by histopathology, but also small cancer foci in areas indicated as fibro-adipose by histopathology. These areas might be real, as the histopathology was

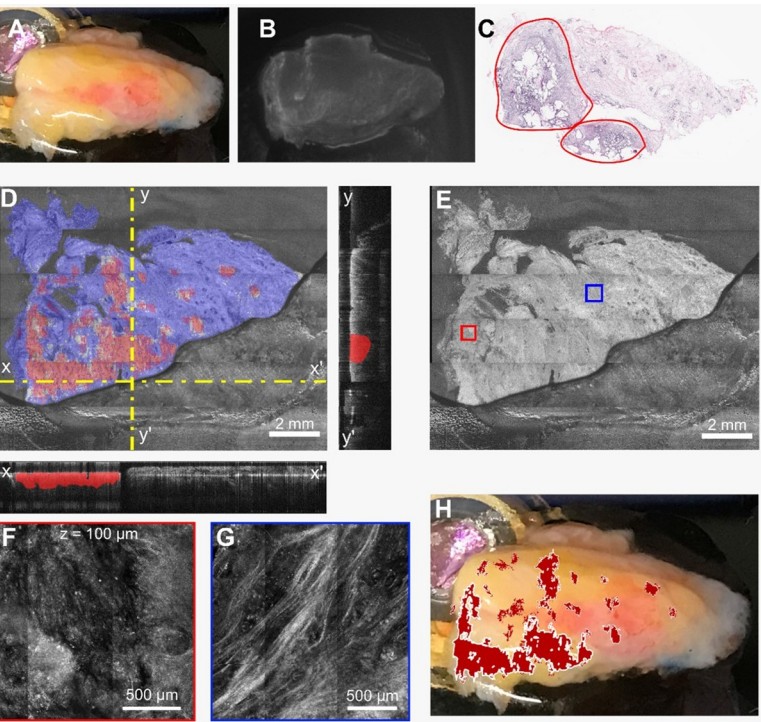

**Fig 12. Example of positive margin identification and potentially false negatives indicated by histopathology on a heterogeneous specimen. A)** Camera photo of the specimen on the microscope stage. **B)** Fluorescence imaging seems to indicate an area of tumor in the top-right corner (yellow). **C)** Annotated histology indicating cancer areas (red annotations). **D)** Enface segmented OCT image with corresponding X and Y cross-sectional images. Dotted-yellow line shows the plane of the cross-sectional images. **E)** En face OCT image. Red and Blue boxes on the OCT image show the regions of RCM images in the F and G panels. **H)** Overlay of the positive margins on the surgical specimen.

unfortunately performed at one single depth, closer to surface, while additional cancer foci might been present at higher depths.

## 4. Discussion

A multimodal instrument was developed with the goal of examining breast surgical resections encountered in routine clinical practice for positive margin presence. This novel instrument combines three optical imaging modalities: fluorescence, confocal, and optical coherence tomography imaging. To enable enhanced contrast fluorescence imaging, a custom contrast agent was developed by attaching Cy5.5 to a quenched uPa vulnerable sequence. Analysis of the cleaved increase in fluorescence correlates with the presence of cancer in the tissue. The contrast agent and fluorescent microscopy gives an initial estimate of residual cancer on the surface of the tissue. However, because evaluation by fluorescence microscopy alone can be subject to non-specific positive margin highlighting, the utility of adding OCT and RCM becomes crucial in margin analysis. The RCM provides submicron detail to inform tissue-type classification, and guide OCT data segmentation, which provides volumetric mapping of the extent of the tumor infiltration. By using the FL mode to identify areas of highly suspicious for presence of cancer, the acquisition time can be greatly reduced during RCM/OCT imaging, which is critical for intraoperative utility. The information from the three modalities was combined in an image-processing regimen to automatically detect the cancer margins in 3D very effectively.

Five cases were highlighted in this paper to demonstrate the potential clinical utility of the multimodal optical imaging for breast cancer surgery guidance. The first case presented showed uniform staining with a near complete increase in FL signal from the contrast agent. RCM/OCT imaging and subsequent image analysis showed similar agreement, that the margins extended to the surface and edges and that the tumor content was distributed throughout the depth. Histopathological analysis confirmed that the specimen was an invasive ductal carcinoma, and the annotated section correlated well with the RCM/OCT images.

The second case was a more complicated specimen and more representative of the type of tissue found at tumor boundaries. Visual inspection showed a clear interface between yellow, likely fat and whiter, potentially malignant tissue. The FL image showed increased brightness from the contrast agent in approximately the same areas as observed visually, though the signal ratio was not as high as in case 1. The RCM/OCT imaging showed a more complicated tissue structure than was seen in case 1. The annotated histology of the hematoxylin and eosin stained tissue section indicated invasive ductal carcinoma amidst normal breast tissue. At certain depths, the histological annotations matched well with the image analysis prediction. However, through the analysis of the volumetric OCT data, we predict the tumor extended laterally in depth. In this study, the specimens were sectioned only a limited number of times. For 3D ground truth from histology, the specimen would need to be sectioned serially throughout, which is costly, introduces delays, and requires lengthy expert interpretation. This emphasizes that further improvements of this instrument is feasible which will be useful for eventual incorporation into surgical pathology practice.

The third presented case is one that visually appears like uniform adipose tissue. The FL channel did not show increased fluorescence signal and the histopathologist indicated no tumor content. Although in real clinical practice additional OCT imaging would be unnecessary when RCM indicates the absence of the positive margins, in this case we performed OCT imaging to confirm that the segmentation algorithm assigns this area to the correct tissue class.

The fourth case is that of a heterogeneous specimen, where multiple tissue types can be visualized by eye. The fluorescent contrast agent showed a small region of potentially positive margins, which warranted more in-depth analysis. However, histological annotation showed the presence of only multiple benign breast lobules and not of cancer. OCT also showed some small false positives and the lobules areas. However, RCM confirmed the presence of benign tissue.

The fifth case showed the utility of OCT depth sectioning capabilities, as hidden positive margin areas can subside underneath the specimen surface, within 1 mm depth. Histology might fail showing these areas if, as usually in the clinical practice, a single slice at a single depth is analyzed.

Histology correlation of the imaging findings was difficult, as perfect correlation of the OCT/RCM images with histology was almost impossible due to alterations of tissue alignment with processing of the tissue during histology preparation. This is particularly apparent in samples with a high content of adipose tissue. For this reason, precise percentages of tissue type area coverage were not computed or compared. Furthermore, the histological practice often outlines large areas that have a mix of tissues within them out of an abundance of caution and to ensure that no tumor cells remain. For example, fat cells within a tumor area, would be segmented as fat cells even though they would be located in a region marked for removal.

The segmentation algorithm correctly classified many complex tissue types as benign, such as fibrosis and treatment-related change from malignant to benign. However, some false positives after OCT image segmentation were still present in two cases. As RCM cannot be performed fast enough on large areas highlighted by FL imaging, this technology needs further improvements to eliminate the false positives. Potentially, larger training sets may improve the

false positive rate in areas of normal lobules, as well as the use of more sophisticated algorithms for tissue-type differentiation. More precise histology alignment and preparation will potentially allow for registration of images and tissue type local area comparisons.

Taken together, this case study suggests the novel multimodal instrument can be of great use in identifying surgical margins on breast cancer resection specimens. As noted, rapid feedback of margin extent and location could reduce the need for repeated surgeries and cosmetic damage.

## 5. Conclusion

Multimodal optical imaging was performed on breast surgical specimens with the goal of evaluating the potential of this technology for assessing surgical margins as a tool for surgery guidance. Twenty specimens were evaluated, and a reasonably good agreement between imaging findings and histopathology findings were not noted. However, overestimation of positive margins was noted on several tissue specimens as compared to histology findings. Although some of the overestimates might indeed be false-positives, it is also possible that they were true-positives, considering perfect histology correlation was impossible due to registration obstacles. Some small false positives were flagged by OCT algorithmic segmentation in areas of normal lobules. RCM analysis correctly rejected these false positives in most of the cases. Histology processing usually distorts the shape of the specimen, particularly in tissues high in adipocytes. In addition, a single histology slide was taken from each surface of the surgical specimen from a depth of approximately 100 um beneath the specimen surface, while RCM and OCT imaging was performed to depths of 100 μm and 1mm, respectively. The deeper, volumetric information sometimes revealed margins that extended beyond the histology indication. Further evaluation of this technology on additional specimens and better correlation with histology is ongoing which will be reported in a follow-up study. Nevertheless, the preliminary findings are encouraging, showing the potential benefits of this technology for intraoperative evaluation of breast surgical specimens.

## 6. Limitations and future work

Despite encouraging preliminary results, improvements on instrumentation, data collection and data analysis will be considered for future studies. Registration of OCT/RCM images and histological must be reconsidered. The specimens here were photographed and marked with a dot of tissue dye to aid in alignment for sectioning. However, there are undoubtedly slight misalignments in pitch and roll rotations. This affects the accuracy of the matching tumor percentages globally and spatially between RCM/OCT and histology. Future studies will attempt to utilize custom cassettes during fixation or imaging and sectioning will be performed using frozen sections within the same laboratory space as soon as possible for better correlation of histology and imaging. The processing algorithm will also need to be based on larger training sets, as there is significant variation within the scattering properties of tissue specimens belonging to the same tissue class, which contributes to the false-positive rate in normal lobule areas.

Further studies will also attempt to analyze the contrast agent in more details in examine potential options for providing more selective cancer binding, as well as to add a fluorescence channel to the confocal system, as fluorescence CM shall improve pathologist ability to differentiate tissue-types, as demonstrated by other studies [31].

This paper presents only limited case studies; further evaluation of the instrument will be performed using more resection specimens. This will greatly improve the image processing

and automated classification techniques. As datasets become larger and more diverse, machine learning techniques can be implemented.

## Author Contributions

**Conceptualization:** Mark T. Scimone, Savitri Krishnamurthy, Dorin Preda, Jesung Park, John Grimble, Nicusor Iftimia.

**Data curation:** Mark T. Scimone, Savitri Krishnamurthy, Dorin Preda, John Grimble, Min Song, Kechen Ban.

**Formal analysis:** Gopi Maguluri.

**Funding acquisition:** Nicusor Iftimia.

**Investigation:** Mark T. Scimone, Savitri Krishnamurthy, Jesung Park, John Grimble, Min Song, Kechen Ban, Nicusor Iftimia.

**Project administration:** Nicusor Iftimia.

**Resources:** Savitri Krishnamurthy.

**Software:** Mark T. Scimone, Gopi Maguluri.

**Supervision:** Nicusor Iftimia.

**Writing – original draft:** Nicusor Iftimia.

**Writing – review & editing:** Mark T. Scimone, Savitri Krishnamurthy, Dorin Preda, Nicusor Iftimia.

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
