## [Decision Letter · Decision Letter 0]

15 Oct 2020

PONE-D-20-28159

Assessment of breast cancer surgical margins with multimodal optical microscopy: a preliminary clinical study

PLOS ONE

Dear Dr. Iftimia,

Thank you for submitting your manuscript to PLOS ONE. After careful consideration, we feel that it has merit but does not fully meet PLOS ONE’s publication criteria as it currently stands. Therefore, we invite you to submit a revised version of the manuscript that addresses the points raised during the review process.

As you will see the two reviewers have some significant issues with the manuscript that need to be addressed. Specifically, the first reviewer brings up a number of points about lack of data on the target specificity, kinetics of probe activation, depth penetrance and analytical characterization. These are concerns that need to be addressed in your revision. The second reviewer mainly had minor comments and questions that are marked up in the pdf file attached. Please address all of these comments as well. I am happy to consider a revision of the work as long as you are able to address all of the reviewer comments with the addition of new data or with a reasonable explanation as to why additional data is not necessary.

We look forward to receiving your revised manuscript.

Kind regards,

Matthew Bogyo, Ph.D.

Academic Editor

PLOS ONE

Journal Requirements:

2. Please ensure your Methods and reagents are be described in sufficient detail for another researcher to reproduce the experiments described. Specifically, please provide further details on the methods of section 2.1 Flourescence imaging agent, including details of conditions for synthesis.

3. Please include the results for the other 10 samples as supplementary information.

4. Please upload your figures as separate files.

5. In the ethics statement in the manuscript and in the online submission form, please provide additional information about the patient samples used in your retrospective study, including: a) the date range (month and year) during which patients' samples records were retrieved; b) the date range (month and year) during which patients whose samples were selected for this study sought treatment. If patients provided informed written consent to have samples from their medical records used in research, please include this information.

6. Thank you for including your ethics statement: 'Breast surgical specimens were provided by MD Anderson Cancer Center. Fourteen deidentified specimens were acquired from breast cancer patients undergoing surgery procedures (both lumpectomy and mastectomy) under the IRB approved protocol PA14-1036.'

a. Please amend your current ethics statement to include the full name of the ethics committee/institutional review board(s) that approved your specific study and confirm that your named institutional review board or ethics committee specifically approved this study.

7.Thank you for stating the following in the Financial Disclosure section:

[Nicusor Iftimia-PI

2R44CA173998-02A1

US National Cancer Institute

The funders had no role in study design, data collection and analysis, decision to publish, or preparation of the manuscript.].   

We note that one or more of the authors are employed by a commercial company: Physical Sciences Inc

Reviewers' comments:

Reviewer's Responses to Questions

**Comments to the Author**

1. Is the manuscript technically sound, and do the data support the conclusions?

Reviewer #1: No

Reviewer #2: Yes

2. Has the statistical analysis been performed appropriately and rigorously? 

Reviewer #1: Yes

Reviewer #2: No

3. Have the authors made all data underlying the findings in their manuscript fully available?

Reviewer #1: No

Reviewer #2: Yes

4. Is the manuscript presented in an intelligible fashion and written in standard English?

Reviewer #1: Yes

Reviewer #2: Yes

5. Review Comments to the Author

Reviewer #1: There are several aspects of this manuscript that are very exciting and have the potential to impact medical management of BCS. This manuscript presents data to demonstrate the utility of the method. However, the manuscript does not present enough information about aspects of the technology to fully interpret its impact. I will review the data presented and then the overall impact of the data.

Data presented:

Overall the authors are assessing a multi-modality optical imaging modality to assess tumor margins in lumpectomy tissues ex vivo. The approach has been developed to identify regions likely to have cancer using FL guided imaging (and a probe to assess tumor presence) and then utilizes other optical approaches to verify tumor presence and depth of tumor lesions (OCT and RCM).

Probe: The description of the probe defines it as a Cy5.5 labeled probe that is self-quenched. The quench is released by uPA protease. The probe is also targeted to Her2 receptors, the sequence of the targeting peptide is withheld from the readers. There is in vitro data to describe the dequenching of probe and the level of activation. Methodology of use is quite straight forward. The probe is simply applied to ex vivo tumor resection samples for 1-2 minutes, rinsed and then FL measured. If FL regions are detected other means of imaging are used to confirm presence of cancer and report on its penetration. The level of un-quenched probe washout after a 2-minute incubation and washing should be assessed. Perhaps specific signal is lost.

There is no data to demonstrate specificity of the probe for Her2 binding, nor protease activation. It would be good for there to be data explaining the tissue characteristics of probe activation. Using protease inhibitors specificity of probe activation could be established. Similarly using peptide competitors for Her2, receptor binding could be assessed. What is the kinetics for Her2 binding to the peptide in tissues samples, and for that matter do in vitro protease activation kinetics represent probe activation kinetics in tissue samples. Using mouse models with topical application of the probe, these could easily be answered.

It is not clear why the probe should penetrate tissue at all given the short duration of incubation. Solubility measures of the probe should be presented and penetration studies into human tumors grown in mice would help inform the statement of 1 mm probe penetration. In general, topical applications of the probe is aqueous carriers do not promote tissue penetration.

Finally, a reasonable discussion of the expression level of uPA in breast cancer tumors should be placed within the paper and include the presence of Her2 receptor as well. Justification for the synthetic design of this probe is sorely needed.

Patient samples: Given the Her2 binding ability of the probe the patient samples used in the study should be identified as being Her2 positive or not. This information is not provided. Her2 only is overexpressed in around 20% of breast cancer patients, so it is not clear if utilizing the Her2 peptide ligand is helpful or not. Her2-positive and Her2-negative samples could be compared to describe the impact of the peptide ligand on tumor tissue recognition.

The paper starts off claiming 100% correlation of the data with ground truth histology but later reveals that the correlation is only in histology verified positive lumpectomy specimens, and that false positives were detected. It is not clear if the multimodality approach was able to resolve these before or after histological verification.

Cancer visualization: It is not clear the sensitivity of the methodology. Many pathological assessments call positive margins with very few cancer cells present. What is the lowest number of cancer cells that can be FL detected? The time for such analysis also needs to be demonstrated. The author states it speeds up analysis, but this is due to reducing the amount of tissue that OCT or Raman are used to study. What is the total average time to sample and analyze the lumpectomy samples for the presences of cancer? Is it faster than frozen sectioning?

The author also states that correlations of FL and OCT images back to fixed tissue is difficult or impossible to perform. This has been solved for skin cancer and the author is referred to Walker et al., Cancer Research 2020.

It is not clear is the cancer specimens were assessed from all sides. To gain true assessment of the utility of this approach all edges and sides of the tumor specimens must be assessed. With that in mind it would be good for the author to present sensitivity and specificity table for the samples analyzed.

Was the study blinded to histology outcomes?

Overall Impact of the Study:

The impact of the approaches demonstrated here could be significant for management of breast conserving surgery. However, many of the parameters that are required for this assessment are missing, e.g. specificity of the probe, Her2 positivity of tested samples, time to perform the procedure, the ability to “see” the entire surface of the lump and stitch together a useful map of cancer at the margin to go back to the surgical cavity and in real time re-excise regions with positive margins. This would have been interesting to discuss in the future perspectives of the discussion section.

Reviewer #2: It would be helpful to implement the attached edits within the pdf and have the manuscript read by a native English speaker.

Overall this a well-conducted study that suffers a bit from a small sample size but highlights how the sum of multiple imaging modalities can make up for deficiencies inherent in any single method.

6. PLOS authors have the option to publish the peer review history of their article (what does this mean?). If published, this will include your full peer review and any attached files.

Reviewer #1: No

Reviewer #2: **Yes: **Jonathan Sorger

---

## [Author Response · Author response to Decision Letter 0]

1 Dec 2020

Reviewer 1:

Q1: The level of un-quenched probe washout after a 2-minute incubation and washing should be assessed. Perhaps specific signal is lost.

Answer: The unquenched probe contributes to the background fluorescence signal, as observed from Fig. 2 (fluorescence background before cleavage- created by the unquenched agent). This level is rather small, as the fluorescence increases 10 fold after peptide cleavage. However, we believe that the reviewer refers to the un-cleaved peptide after 2 minutes of incubation (see plot from Fig. 3 - revised manuscript). This level can vary as a function of the enzyme-substrate ratio. The more enzyme on the tissue, the faster is the cleavage -see E:S = 1:25 vs. E:S= 1:200. We noticed a similar behavior during the experiments performed on tissue specimens. Since not all specimens express the same level of uPA enzyme, the level of un-cleaved peptide varies after 2-minute of incubation. To determine the amount of un-cleaved peptide left after incubation, we added enzyme to the washout solution from several specimens and noticed no fluorescence increase in some cases and up to 20% fluorescence increase, confirming that the uPA concentration varies from one specimen to another. For specimens that express lower uPA levels, longer incubation provides improved cleavage. However, since small concentrations of uPA are present as well in the surrounding benign fibrotic tissue, longer incubation times actually result in a decrease of the fluorescence contrast of the tumor tissue. Based on these facts, we determined that 2 minutes of incubation provides optimal fluorescence contrast. 

Q 2. There is no data to demonstrate specificity of the probe for Her2 binding, nor protease activation. It would be good for there to be data explaining the tissue characteristics of probe activation. Using protease inhibitors specificity of probe activation could be established. Similarly using peptide competitors for Her2, receptor binding could be assessed. What is the kinetics for Her2 binding to the peptide in tissues samples, and for that matter do in vitro protease activation kinetics represent probe activation kinetics in tissue samples. Using mouse models with topical application of the probe, these could easily be answered.

Answer: Unfortunately, our funded NIH grant does not support animal studies, and therefore this experiment cannot be performed as suggested. Furthermore, it was not the focus of this research to study the effectiveness of the contrast agent in tagging the Her2 receptor. This would warrant a full separate study, as it was not the intent of our current study. Therefore, we removed from the manuscript any discussion related to this subject. We have changed the manuscript as follows: “A spacer (aminohexanoic acid) may be used to ensure that this peptide moiety will maintain fast uPA cleavage kinetics. However, enhancement of cancer specificity was not a major focus of this preliminary study.”

Q3. It is not clear why the probe should penetrate tissue at all given the short duration of incubation. Solubility measures of the probe should be presented and penetration studies into human tumors grown in mice would help inform the statement of 1 mm probe penetration. In general, topical applications of the probe is aqueous carriers do not promote tissue penetration.

Answer: We agree with the reviewer that 2 minutes of incubation might not provide very deep tissue penetration. However, 1 mm penetration is sufficient, as at least 1 mm safe margin is needed in surgery. Based on our experiments, after 2 minutes of contrast agent placement on the top surface (pipette drops), the opposite side on a specimen with 1 mm thickness showed similar fluorescence on both surfaces. However, as suggested, further experiments will be performed in a subsequent study to address further increase in solubility measures.

Therefore, the manuscript was modified as follows: “This suggests that that the optimum incubation time will be about 1 minute. However, to enable contrast agent diffusion in the tissue, our experiments showed that a maximum of 2 minutes incubation time is a good compromise between fluorescence contrast and agent diffusivity to about 1 mm deep. For longer incubation times, the smaller concentrations of the uPA enzyme present on the benign tissue start cleaving the agent, and thus degrade tumor contrast.”

Q4. Finally, a reasonable discussion of the expression level of uPA in breast cancer tumors should be placed within the paper and include the presence of Her2 receptor as well. Justification for the synthetic design of this probe is sorely needed.

Answer: The expression level of the uPA in cancer tumors is discussed in the paper: "Based on literature data, we estimated the concentration of the enzyme that is present in malignant tissue versus benign tissue, and used these numbers as a baseline for our measurements. Literature reports (30) show that the average uPA level in benign tissue is 1 ng/mg protein, while in tumor tissue is 8 ng/mg protein." The synthetic design of the probe is not the main goal of this paper, as in our study we focused on the microscopy aspects. Any other contrast agents can be used with this microscope. Therefore, we modified the manuscript as follows: “ A contrast agent capable of highlighting areas of the specimen that have potential positive margins can be used with the proposed microscope. The main requirement is to have close to 100% sensitivity, as otherwise positive margins may be left behind. For this study, a novel contrast agent was developed by our team. It consists of an optically silent peptide substrate containing two (near-infrared; NIR) fluorochromes (Cy5.5), internally quenched. This agent was synthesized to track positive margins on breast lumpectomy specimens via specific urokinase plasminogen activator (uPA) cleavage.”

Q5. Patient samples: Given the Her2 binding ability of the probe the patient samples used in the study should be identified as being Her2 positive or not. This information is not provided. Her2 only is overexpressed in around 20% of breast cancer patients, so it is not clear if utilizing the Her2 peptide ligand is helpful or not. Her2-positive and Her2-negative samples could be compared to describe the impact of the peptide ligand on tumor tissue recognition.

Answer: Her-2 positive and Her-2 negative specimens cannot be easily identified and thus such experiments were beyond the scope of this paper. HER2 testing through immunohistochemistry and FISH has shown error rates as high as 20%. A key aspect of the presented technology is that other contrast agents may be used if necessary or preferred. 

Q6. The paper starts off claiming 100% correlation of the data with ground truth histology but later reveals that the correlation is only in histology verified positive lumpectomy specimens, and that false positives were detected. It is not clear if the multimodality approach was able to resolve these before or after histological verification.

Answer: From the presented data it is clear that the assessment of multimodal imaging effectiveness was performed by comparing imaging findings against histopathology findings. The imaging study was first performed and then the imaging results were compared against histopathology results. A table was inserted to show the summary of the findings. 

Table 1: Summary of imaging and histopathology findings

20 Specimen study Fluorescence imaging RCM guided OCT segmentation Histopathology

Positive margins specimens 12 10 (2FP) 8 TP

Negative margins specimens 8 10 (0FN) 12 TN

RCM-guided OCT predictive values: PPV=1.0; NPV= 0.83

Q7. Cancer visualization: It is not clear the sensitivity of the methodology. Many pathological assessments call positive margins with very few cancer cells present. What is the lowest number of cancer cells that can be FL detected? The time for such analysis also needs to be demonstrated. The author states it speeds up analysis, but this is due to reducing the amount of tissue that OCT or Raman are used to study. What is the total average time to sample and analyze the lumpectomy samples for the presences of cancer? Is it faster than frozen sectioning?

Answer: Epifluorescence imaging cannot be used visualize individual cancer cells, as its resolution is ~ 50 um. Therefore, only clusters of cells can be visualized. Single cancer cell detection is out of the question, and not really important, as the usual clinical protocol is to have a boost radiation of the surgical bed and destroy eventual small clusters of cells. Indeed, fluorescence imaging reduces the time for microscopy analysis. The analysis time varies as a function of the size of the fluorescence-indicated margin. In our study, this area varied between 25 mm2 and 200 mm2. With an OCT frame rate of 40 frames/sec (40 B-scans/sec), a B-scan size of 2.2 mm, and a 20 um separation between B-scans, an area of 100 mm2 requires ~2 minutes. Again, frozen sectioning is not suitable for breast specimens, as they have a high adipose content and are thus very difficult to section. 

Q8. The author also states that correlations of FL and OCT images back to fixed tissue is difficult or impossible to perform. This has been solved for skin cancer and the author is referred to Walker et al., Cancer Research 2020.

Answer: The imaged specimens were marked with ink on the top surface and histology processing was carefully performed to correlate the correct side with imaging results. However, when we refer to imaging-histology correlation, we consider the fact that when fixing the tissue, it shrinks and thus the histology images of the specimen do not look exactly like the enface OCT images, and thus 100% correlation of the findings is not really possible. Furthermore, skin cancer and epidermal/dermal tissue maintain fidelity during sectioning much better than tissues high in adipose. Difficulties surrounding sectioning of adipose tissue include: adipose tissue is more prone to ripping, requires lower temperatures for frozen sectioning, and requires longer delay times for embedding (in paraffin or OCT) to ensure the embedding material supports the structure.

Q9. It is not clear is the cancer specimens were assessed from all sides. To gain true assessment of the utility of this approach all edges and sides of the tumor specimens must be assessed. With that in mind it would be good for the author to present sensitivity and specificity table for the samples analyzed.

R: The tissue specimens were relatively thin (~ 3-4 mm when flattened within the instrument specimen holder). Therefore, they were analyzed on the two opposite surfaces. A table showing the positive and negative predictive values from a limited set of specimens was inserted within the revised manuscript. Since this was a feasibility study on a small set of specimens, a clear assessment of technology sensitivity and specificity might be premature.

Q10. Was the study blinded to histology outcomes?

Answer: Yes, the microscopy data were first analyzed and then tested against the histopathology reports. 

Q11. Overall Impact of the Study:

The impact of the approaches demonstrated here could be significant for management of breast conserving surgery. However, many of the parameters that are required for this assessment are missing, e.g. specificity of the probe, Her2 positivity of tested samples, time to perform the procedure, the ability to “see” the entire surface of the lump and stitch together a useful map of cancer at the margin to go back to the surgical cavity and in real time re-excise regions with positive margins. This would have been interesting to discuss in the future perspectives of the discussion section.

Answer. We agree that a more in-depth study will be needed to further analyze these aspects. However the main goal of this paper was to report the potential impact of this technology. The conclusion section has been amended with the next paragraph: “Further studies will also attempt to analyze the contrast agent in more details in examine potential options for providing more selective cancer binding. This paper presents only limited case studies; further evaluation of the instrument will be performed using more resection specimens. This will greatly improve the image processing and automated classification techniques. As datasets become larger and more diverse, machine learning techniques can be implemented. ”

Reviewer 2:

1. It would be helpful to implement the attached edits within the pdf and have the manuscript read by a native English speaker.

2. Answer: English speaker proofreading was performed. We thank the reviewer for their careful proofreading and suggestions.

2. Overall this a well-conducted study that suffers a bit from a small sample size but highlights how the sum of multiple imaging modalities can make up for deficiencies inherent in any single method.

Answer: We agree with the reviewer that this was a relatively small sample size study. We included an additional of 6 cases and a table indicating technology performance.

---

## [Decision Letter · Decision Letter 1]

29 Dec 2020

Assessment of breast cancer surgical margins with multimodal optical microscopy: a feasibility clinical study

PONE-D-20-28159R1

Dear Dr. Iftimia,

We’re pleased to inform you that your manuscript has been judged scientifically suitable for publication and will be formally accepted for publication once it meets all outstanding technical requirements.

Kind regards,

Matthew Bogyo, Ph.D.

Academic Editor

PLOS ONE

Additional Editor Comments (optional):

Reviewers' comments:

Reviewer's Responses to Questions

**Comments to the Author**

1. If the authors have adequately addressed your comments raised in a previous round of review and you feel that this manuscript is now acceptable for publication, you may indicate that here to bypass the “Comments to the Author” section, enter your conflict of interest statement in the “Confidential to Editor” section, and submit your "Accept" recommendation.

Reviewer #1: All comments have been addressed

Reviewer #2: All comments have been addressed

2. Is the manuscript technically sound, and do the data support the conclusions?

Reviewer #1: Yes

Reviewer #2: Yes

3. Has the statistical analysis been performed appropriately and rigorously? 

Reviewer #1: Yes

Reviewer #2: N/A

4. Have the authors made all data underlying the findings in their manuscript fully available?

Reviewer #1: Yes

Reviewer #2: Yes

5. Is the manuscript presented in an intelligible fashion and written in standard English?

Reviewer #1: Yes

Reviewer #2: Yes

6. Review Comments to the Author

Reviewer #1: (No Response)

Reviewer #2: (No Response)

7. PLOS authors have the option to publish the peer review history of their article (what does this mean?). If published, this will include your full peer review and any attached files.

Reviewer #1: No

Reviewer #2: No

---

## [Editor Report · Acceptance letter]

29 Jan 2021

PONE-D-20-28159R1 

Assessment of breast cancer surgical margins with multimodal optical microscopy: A feasibility clinical study 

Dear Dr. Iftimia:

I'm pleased to inform you that your manuscript has been deemed suitable for publication in PLOS ONE. Congratulations! Your manuscript is now with our production department. 

Kind regards, 

on behalf of

Dr. Matthew Bogyo 

Academic Editor

PLOS ONE